# Effect of Low Light Stress on Distribution of Auxin (Indole-3-acetic Acid) between Shoot and Roots and Development of Lateral Roots in Barley Plants

**DOI:** 10.3390/biology12060787

**Published:** 2023-05-29

**Authors:** Alla Korobova, Ruslan Ivanov, Leila Timergalina, Lidiya Vysotskaya, Tatiana Nuzhnaya, Guzel Akhiyarova, Victor Kusnetsov, Dmitry Veselov, Guzel Kudoyarova

**Affiliations:** 1Ufa Institute of Biology, Ufa Federal Research Centre RAS, 69 Pr. Octyabrya, 450054 Ufa, Russia; muksin@mail.ru (A.K.); ivanovirs@mail.ru (R.I.); leinaz@mail.ru (L.T.); vysotskaya@anrb.ru (L.V.); tanyawww89@mail.ru (T.N.); akhiyarova@rambler.ru (G.A.); guzel@anrb.ru (G.K.); 2K.A. Timiryazev Institute of Plant Physiology RAS, 35 Botanicheskaya St., 127276 Moscow, Russia

**Keywords:** *Hordeum vulgare* L., low light stress, auxin (IAA), phloem, root branching, *AUX1*, *LAX3* genes

## Abstract

**Simple Summary:**

Depending on their habitat conditions, plants can greatly change the patterns of biomass allocation between different organs. Thus, low light inhibits the growth of roots, but at the same time, plants maintain elongation of their shoots. Phytohormones ensure the coordination of processes occurring in different plant organs due to the transmission of signals regarding changes in the environment. They are synthesized both in roots and shoots. Their transport from roots to shoots and vice versa occurs through the xylem and phloem, respectively. It is believed that long distance transport of auxins is important for the control of the growth of roots under conditions of low light; however, the mechanism of these effects remains unclear. The inhibition of lateral root emergence in barley under reduced levels of irradiance is due to a disturbance of auxin transport through the phloem and down-regulation of the genes responsible for auxin transport in plant roots. A comparison of results obtained for wheat and barley shows that mechanisms controlling the auxin signal sent from shoots to roots can be different and species specific.

**Abstract:**

Depending on their habitat conditions, plants can greatly change the growth rate of their roots. However, the mechanisms of such responses remain insufficiently clear. The influence of a low level of illumination on the content of endogenous auxins, their localization in leaves and transport from shoots to roots were studied and related to the lateral root branching of barley plants. Following two days’ reduction in illumination, a 10-fold reduction in the emergence of lateral roots was found. Auxin (IAA, indole-3-acetic acid) content decreased by 84% in roots and by 30% in shoots, and immunolocalization revealed lowered IAA levels in phloem cells of leaf sections. The reduced content of IAA found in the plants under low light suggests an inhibition of production of this hormone under these conditions. At the same time, two-fold downregulation of the *LAX3* gene expression, facilitating IAA influx into the cells, was detected in the roots, as well as a decline in auxin diffusion from shoots through the phloem by about 60%. It was suggested that the reduced emergence of lateral roots in barley under a low level of illumination was due to a disturbance of auxin transport through the phloem and down-regulation of the genes responsible for auxin transport in plant roots. The results confirm the importance of the long distance transport of auxins for the control of the growth of roots under conditions of low light. Further study of the mechanisms that control the transport of auxins from shoots to roots in other plant species is required.

## 1. Introduction

Unlike animals, plants can greatly change their patterns of biomass allocation between different organs depending on their habitat conditions, which ensures their adaptation to the availability of resources. Thus, a deficit of water and mineral nutrients inhibits the growth of shoots, but at the same time, plants maintain elongation and branching of their roots, which is necessary for the acquisition of resources in the soil [1]. Conversely, low light activates shoot and leaf growth, allowing leaves to be exposed to higher light intensity in the upper canopy layers and increasing leaf area to improve the efficiency of light interception [2]. Activation of the growth of some organs often occurs due to the release of resources as a result of the inhibition of the growth of other organs. Inhibition of shoot growth by a deficiency of mineral nutrients and water releases resources for root growth, while inhibition of development of root system in low light conditions saves the resources necessary for shoot growth. For example, it has been shown that low illumination inhibits roots branching [3]. Such responses suggest the existence of mechanisms that ensure the coordination of processes occurring in different plant organs due to the transmission of signals regarding changes in the environment. It is believed that phytohormones play the role of such signal transmitters. They are synthesized either in roots or in shoots, and their transport from roots to shoots and vice versa occurs through the xylem and phloem, respectively. Changes in the living conditions of plants affect hormone metabolism, which leads to a change in the flow of hormones from organ to organ, thereby transmitting information about local external influences. As a result, plant reaction is realized at the level of the whole organism due to the ability of hormones to control various processes. The mechanism of signal transduction from roots to shoots has been well studied using the example of the effect of drought on ABA (abscisic acid) synthesis in drying roots. It has been shown that the increase in its flow to the shoot contributes to stomatal closure, reduced transpiration and shoots growth [4]. Signaling in the opposite direction from shoots to roots received less attention, which could be due to the difficulty in assessing the hormone content in phloem sap. Nevertheless, a decrease in root branching under the influence of low light was explained by a decrease in auxin transport from shoots to roots [5]. At the same time, these data, based on the assessment of the ability of exogenous auxins to stimulate branching under low light, did not always give unambiguous results. Although low concentrations of the exogenous hormone did stimulate branching, high concentrations of auxin inhibited the development of lateral roots [6]. Obviously, the assessment of the effect of illumination on the content of endogenous auxin (IAA) should provide more realistic information. At the same time, there are very few works that have studied the effect of illumination on the level of IAA in connection with its transport to the roots and the effect on root branching. The data on the effects of low light on auxins are rather contradictory. For example, a study of early responses to shade signals showed that auxin synthesis and transport are rapidly activated in *Arabidopsis*, thus promoting cell elongation of hypocotyls [7]. In contrast, other researchers have found greater auxin content in the shoots of *Arabidopsis* plants at high compared to low light [2].

Previously, we revealed a decrease in the delivery of IAA to the roots with a decrease in the illumination of wheat plants [8]. Attempts to associate this phenomenon with the effects of light intensity on the expression of auxin transporters in the shoot, where IAA is loaded into the phloem, did not reveal any changes in the level of transcripts of the *PIN* gene [8], encoding the efflux carrier of auxins [9]. It has been suggested that the decrease in IAA delivery from shoots to roots under the influence of low light could be due to the expression of other auxin transporter genes. In the present work, we studied the effect of low light on the expression of genes encoding another class of carriers (AUX1/LAX) that transport auxins in the opposite direction (into cells). Expression studies on *Arabidopsis* plants using IAA::uidA reporter revealed that AUX1 contributes to auxin loading into the leaf phloem [10]. Despite these promising data, most researchers have studied the role of root-expressed transporter genes in controlling lateral root development [11]. It is easy to see that most of cited papers report the results of experiments with *Arabidopsis* (excluding the work of Reed and coauthors, which was carried out on maize plants [5]), while in the case of cereals, one may expect different responses. Barley plants were chosen by us for the present research, since the genome of these plants is better studied than that of wheat plants, which facilitated the search for genes of interest. Barley is the fourth most important cereal crop after corn, rice and wheat in terms of production quantity. It is the predominant crop in some dryland agricultural systems (e.g., in Mediterranean area) [12].

According to some data, AUX1 is required for auxin transport over long distances from shoot to roots through the phloem, while LAX is implicated in maintaining local auxin gradients [13]. In this work, we not only studied the concentration of auxins in the phloem exudate, shoots and roots of plants, but also their distribution between leaf cells, with an emphasis on the auxin level in the phloem and the relationship with the control of lateral root formation. The aim of this research was to reveal the involvement of auxin in the transfer of signals from shoot to root under low light, the mechanisms of the generation of this hormonal signal and its importance for the control of root branching. The mechanisms that ensure the exchange of signals between various plant organs are important for coordinating their growth and adaptation to a changing environment, and a deeper understanding of these mechanisms should allow us to influence these processes in the direction necessary to increase plant productivity.

## 2. Materials and Methods

Seeds of barley plants (*Hordeum vulgare* L., cv. Prairie) were sterilized in 2% sodium hypochlorite solution for 10 min and stratified at 4 °C for 24 h. Then, the seeds were spread on glass rafts floating on the top of Hoagland Arnon’s nutrient solution and were placed in an MLR-351H climate chamber, Sanyo, Japan. The seedlings were grown for two days at an illumination of 165 µmol m^–2^ s^–1^ of photosynthetically active radiation (PAR), temperature of 17 °C/23 °C (night/day) and 16 h photoperiod, and then half of the plants were transferred to a climate chamber with low illumination at 45 µmol m^–2^ s^–1^ PAR (other growth conditions remained the same). One day after the change in illumination, samples of shoots and roots were taken for the quantitative determination of IAA, ABA and RNA isolation. The next day, morphometric parameters were evaluated, including the average total length of all primary roots and the number of lateral roots formed on them. 

**Extraction of RNA and analysis of the transcript level of the genes encoding IAA carriers.** Isolation of total RNA from 7-day-old barley control plants, grown at a relatively high illumination at 165 μmol m^−2^ s^−1^, and experimental plants, grown for one day at 45 μmol m^−2^ s^−1^, was carried out using the “Trizol” reagent according to the recommendation of the producers (Sigma, Steinheim, Germany). The optical density of the sample was measured using the Smart Spec Plus spectrophotometer (Bio-Rad Laboratories, Hercules, CA, USA) as the ratio of absorbance at 260 nm and 280 nm and the concentration of nucleic acids was estimated. For cDNA synthesis, reverse transcriptase M-MuLV (Sintol, Moscow, Russia) was used. Analysis of abundance of the transcripts of *AUX1* and *LAX3* genes was performed by real-time quantitative PCR on a QuantStudio™5 Real-Time PCR System by Thermo Fisher Scientific (Walthan, MA, USA) using EVA Green I intercalating dye (Synthol, Moscow, Russia). The quantitative PCR protocol was as follows: 95 °C for 5 min; 40 cycles at 95 °C for 15 s, at 60 °C for 20 s and at 72 °C for 30 s. The primer set described in Table 1 was used in the work. Barley housekeeping genes encoding Alpha tubulin-2B (*HvTubA*), actin (*HvACT*) and glutamate dehydrogenase (*HvGADPH*) were used as internal controls. Changes in the expression of the genes of interest were normalized using the software CFX Connect real-time PCR Detection System (BioRad Laboratories, Hercules, CA, USA). The measurements were carried out in three chemical and biological replicates. The values obtained for a higher level of illumination are taken as 100%, relative to which, the indicators for low illumination are presented.

**Hormone purification**. Purification of IAA and ABA in shoots and roots was performed as described [14,15]. Contents of hormones were immunoassayed after three days of exposure to low light. Hormones were extracted from crushed shoot and roots with 80% ethanol. After evaporation of the alcohol, aqueous residue was centrifuged and supernatant was collected for further purification. Hormones were purified according to a modified scheme based on a decrease in the volumes of extractants at each stage, which ensured high selectivity of hormone recovery. In short, the pH of aqueous solution was adjusted to 2.5 with HCl and it was partitioned twice with diethyl ether, the ratio of organic to aqueous phases being 1:3. Then, the IAA and ABA were transferred from diethyl ether into sodium hydrocarbonate, the ratio of aqueous/organic phase being 1:3. At the next stage, the pH was readjusted to 2.5 and hormones were reextracted with ether. Reduction of extractant volumes at each stage enabled selectivity of hormone recovery, while it was not less than 80%. After drying, the IAA and ABA were methylated with diazomethane. 

**Hormone quantification**. Hormones were immunoassayed with appropriate specific antibodies. Enzyme Linked Immunosorbent Assay (ELISA) was performed according to competitive protocol [16,17]. In short, conjugate of hormone and ovalbumin dissolved in phosphate buffer was adsorbed onto the walls of microplate wells. After washing the plate thrice with 100 mM NaCl solution containing phosphate buffer, Tween 20 and ovalbumin, a mixture of different concentrations of hormone standard or sample plus anti-hormonal sera were incubated in each well. The principle of the method is based on competition for antibodies between hormones of the sample and a conjugate of hormones and protein absorbed on the walls of the sorbent. After washing unbound rabbit serum, secondary anti-rabbit immunoglobulins, conjugated to peroxidase, were incubated in the wells and, after the next washing, the substrate (o-phenylene-diamine in phosphate buffer pH 5.5 with 3% H_2_O_2_) was added. Light absorption by the product of reaction was measured with a microphotometer (Uniplan, Moscow, Russia).

One day after the transfer of the plants to low light conditions, diffusate from cut shoots was collected in darkness into 5 mM solution of Na_2_EDTA, which prevented the formation of plugs in the phloem [18]. After 15 min, the first portion of exudate was discarded. EDTA chelated the Ca ions which otherwise contribute to sealing the cut phloem [19].

**Auxin immunolocalization**. Immune-histochemical detection of IAA was carried out as described [20]. Leaf and root pieces were fixed in 4% ethylcarbodiimide hydrochloride dissolved in buffered 100 mM solution of NaCl (Merck, Darmstadt, Germany) and then, in 4% paraformaldehyde (Riedel de Haen, Seelze, Germany) and 0.1% glutaraldehyde (Sigma, Steinheim, Germany). Fixed leaf pieces were washed thrice and dehydrated in increasing ethanol concentrations. Then they were embedded in JB-4 plastic resin (Electron Microscopy Sciences, Hatfield, PA, USA). Sections 1.5 μm thick were obtained with microtome HM 325, (MICROM Laborgerate, Walldorf, Germany). Blocking solution containing gelatin and Tween-20 was applied for 30 min (PBS). Then, leaf sections were treated with the rabbit anti-IAA sera washed thrice with solution containing Tween-20 and incubated with secondary antibodies labeled with Alexa Fluor 555 (Invitrogen, Rockford, IL, USA). The slices were washed five times and covered with glass, and images were taken with confocal microscope FV3000 Fluoview (FV31-HSD) (Olympus, Tokyo, Japan) at an excitation of 561 nm and a fluorescence detection of 568 nm.

**Statistical analysis**. The data were statistically processed using Statistica version 10 software (Statsoft, Moscow, Russia). Means and their standard errors are presented on the figures. The significantly different means were revealed by ANOVA followed by Duncan’s test (*p* ≤ 0.05).

## 3. Results

Since our research was focused on the study of the mechanisms that control the growth response of roots to low light, at the first stage, the effects of low light on root growth associated with shoot growth responses were elucidated. Two days after the decrease in illumination, the primary roots of plants became 35% shorter than in plants that grew at a higher level of illumination (control), and the number of lateral roots decreased ten times (2.8 vs. 28; Table 2). The experiments were set up in such a way that at the time of the decrease in illumination, plants did not yet have lateral roots.

To reveal the involvement of hormones in the control of root growth responses to low light, the concentrations of IAA and ABA were determined at the next stage. The content of IAA in the shoots decreased under the influence of low illumination by 35%, and in the roots, almost two times compared to the control. The sharper decrease in the content of IAA in the roots compared to the shoot led to a decrease in the ratio of auxin in the roots and shoots from 1.2 in the control to 0.9 in the experiment (under low illumination). Reduced illumination resulted in a significant decline in the content of IAA in the phloem exudate by 37% compared to the control (Figure 1B). At the same time, this external factor did not significantly affect the content of ABA in the shoots and roots of plants, and according to this indicator, the low-light plants were at the control level (Table 3).

When the sections were treated with antibodies against IAA, immunohistochemical localization revealed a decrease in fluorescence at the site of the phloem tissue on sections of plant leaves transferred to low light conditions (compared to control) (Figure 2). Under low light, fluorescence was present in fewer cells of the phloem than in the control. The fluorescence of phloem cells was mainly coded in green, indicating a low level of auxin content, while in the control, a red and blue color indicated higher auxin content according to the heat map.

There were no differences in the fluorescence of root promordia between plants grown at different illumination levels (Appendix A). The small supply of IAA to the roots under low illumination was apparently still sufficient to provide the reduced number of lateral roots with auxin.

Abundance of the transcripts of the genes encoding IAA transporters was analyzed in order to identify their possible contribution to the control of the hormone distribution between the shoot and root under changing light levels (Figure 3). From our point of view, shoots, where IAA is loaded into the phloem, were of the greatest interest. Since in the previous experiments with wheat, the expression of genes encoding PIP carriers responsible for auxin efflux [9] was not changed in shoots under low light [8], we focused on *AUX1* and *LAX3* genes encoding another class of carriers that transport auxins in the opposite direction (into cells) [10].

However, as in the case of PIN carriers, which were previously analyzed in wheat plants [8], no significant differences were found between the low-light and high-light plants in terms of *AUX1* and *LAX3* gene transcripts in the shoots (Figure 3). At the same time, as in the case of the *PIN* gene, the abundance of *LAX* transcript was lower in the roots of plants grown under low light conditions compared to the control.

## 4. Discussion

In the present experiments, the decrease in IAA concentration in the shoots, roots and phloem found in barley plants under low illumination can be explained by inhibition of auxin synthesis in the shoots and its transport to the roots. Although root tips of *Arabidopsis thaliana* can synthesize auxin, the main sites of auxin synthesis are located in the young leaves and the tips of stems [21]. Therefore, changes in the production of IAA in leaves and its delivery to roots are the most likely candidates for the role of a long-distance signal sent from shoots to roots. However, there is a problem with the interpretation of our data. The problem is that some experiments have shown the opposite reaction in plants—not a decrease, but an increase in synthesis of auxin in the leaves of shaded plants. Thus, in shaded *Arabidopsis* cotyledons, auxin is quickly synthesized through the tryptophan aminotransferase pathway and is transported to the hypocotyl, causing its elongation (a typical shade avoidance response) [22]. An explanation of the controversy between the significance of either increasing or decreasing auxin synthesis for low light response can be found in the article by Hersch et al. [2]. They suggested that the response of plants to an increased ratio of far red (FR) to red (R) in the light spectrum may differ from responses to low light intensity. It has been shown that an increase in the FR/R ratio, which mimics the results of shading due to the presence of neighbors, increases the content of auxins under both low and high light, but the level of this hormone is still lower in low light, which is consistent with the results of the present experiments. It was suggested that the decline in auxin under low light conditions is associated with a decrease in the activity of photosynthesis and production of sugars [2], whereas the metabolism of IAA is influenced by the availability of free sugars [23]. This view was supported by the data showing that supplemented sugar increases auxin concentration and its transport to roots in *Arabidopsis* [24]. Therefore, the decrease in auxin content found in our experiments is likely to be due to the decline in the level of assimilates under low light. In addition, it was shown that increased sensitivity to auxin, rather than enhanced auxin biosynthesis, plays a greater role in cell elongation under prolonged shading [7].

The study of the effects of low light on wheat plants showed that the decrease in root IAA coincided with its accumulation in shoots [8], which can be explained by the inhibition of rootward auxin transport. In the present experiments with barley grown under low illumination, the content of auxins decreased both in shoots and roots, indicating a decrease in the synthesis or an acceleration of the breakdown of auxins under the influence of low illumination. However, changes in the transport of auxin as a mechanism affecting distribution of this hormone between the shoot and root cannot be ruled out, since the degree of decrease in the IAA level under low illumination was higher in the roots than in the shoots. These results indicate a decrease in the loading of auxin into the phloem vessels under the influence of a decrease in illumination, which corresponds to the data on a reduced content of IAA in the phloem exudate of barley plants. The results obtained in this work are consistent with the previous reports that a decrease in illumination in thickened crops leads to a reduced level of auxin in the roots [16]. In these experiments, the use of an inhibitor of auxin transport (naphthylphthalamic acid, NPA) imitated the effects of shading and thereby confirmed that reduced level of root auxin in the presence of neighbors was associated with a decrease in the hormone transport from the shoot.

Experiments on *Arabidopsis* showed that AUX1 (auxin influx transporters) facilitate loading of IAA into the leaf vascular transport system [10]. However, neither mRNA for genes coding for AUX1, nor its analog LAX were up-regulated in the present experiments. As in the case of auxin synthesis, the decreased phloem transport of this hormone may be due to a reduced concentration of photo-assimilates in the plants under low light. According to Munch’s hypothesis confirmed by recent experiments [25], phloem transport is driven by the accumulation of osmotically active substances in the phloem of source leaves, resulting in attraction of water into the phloem from neighboring cells. As a result, pressure is created that drives fluid through the phloem tubes to the roots and other “sink” tissues. Therefore, reduced sugar production under low light should decrease the rate of auxin transport to the roots through the phloem.

During the two days of exposure to low light prior to sampling, the plants had sufficient time to develop lateral roots (according to the available data, this takes about 1.5 days [26,27]). As a result, about 30 lateral roots emerged in the control plants (Table 2). Reduced illumination dramatically slowed down the process of their emergence. The results obtained indicate that the inhibition of the emergence of lateral roots under these conditions was associated with signal regulation. A decrease in the loading and transport of auxins led to a decrease in the level of auxins in the roots, which was accompanied and was the reason for inhibition of the growth of lateral roots. Our results confirm the validity of the assumption that the process of lateral root growth depends on the supply of auxins from the shoot [28]. Removal of the shoot from rice seedlings reduced the density of lateral roots, and the application of IAA to the cut stem restored the lateral root density [29]. It would be interesting to know whether applying IAA to barley plants under low light can restore root branching under low light, which could be a subject for further study.

The inhibition of root branching under the influence of high concentrations of exogenous auxins could be a consequence of the stimulation of ABA synthesis under the influence of high concentrations of auxins. It is known that exogenous auxins can increase the expression of the *NCED* gene, which controls ABA synthesis [30], and ABA, in turn, can inhibit root branching [31]. It has been shown that an increase in the light intensity stimulates accumulation of ABA in the shoots and leaves of wheat plants [32]. However, the effect was transient in plant roots and the difference between low-light and high-light plants in root ABA disappeared with time. In the present experiments with barley plants, a decrease in illumination did not lead to the accumulation of ABA. Therefore, the inhibition of root branching can only be explained by a decrease in the level of auxins in the roots.

AUX1/LAX family members are the major auxin influx carriers, whereas AUX1 and LAX3 both regulate lateral root development (Swarup et al., 2008). [11]. Lateral root development has been shown to be dependent on the local expression of *LAX* genes around lateral root primordial, which facilitates lateral root emergence in *Arabidopsis* [33]. In this case, induction of the *LAX3* gene increased auxins in the root cortex cells, which lead to the up-regulation of cell wall remodeling enzymes to facilitate passage of primordia through the cortex. Therefore, the down-regulation of homologous barley gene found in our experiments in roots under low light may be the cause of the inhibition of lateral root growth and is consistent with some literature data [34]. In the reported experiments, perception of low R:FR regulates abundance of *LAX* and *PIN* in the plasma membranes to decrease auxin levels in the overlaying cortex cells, thereby reducing lateral root outgrowth. It can be assumed that a decrease in gene expression contributed to a decrease in auxin uptake by root cells, which led to the inhibition of root branching. At the same time, an alternative explanation of the results obtained is possible. Thus, it was shown that expression of *LAX* genes, in turn, can be controlled by auxins [33].

The decrease in concentration of auxins in the phloem of barley leaf exudates found in the present experiments under low light was mainly associated with the reduced production of this hormone. These results differ from those obtained on wheat plants, where direct inhibition of auxin transport from shoots to roots led to reduced concentration of this hormone in the roots, but even to an increase in its concentration in the shoots [8]. Thus, a comparison of the results obtained for wheat and barley shows that mechanisms controlling the auxin signal sent from shoots to roots can be different and species specific (mainly due to the inhibition of auxin synthesis in leaves of barley and its transport in wheat). However, despite the difference in the mechanisms of generation of auxin signal, changes in the delivery of this hormone from shoots to roots is apparently responsible for the inhibition of root branching under low light.

## 5. Conclusions

The data obtained in the present work indicate a decrease in the concentration of IAA in the shoots, leaves and phloem sap of barley plants exposed to low light intensity, which suggests a decrease in the synthesis of auxin in the leaves under the influence of this external factor. A decrease in auxin concentration, detected by immunolocalization in leaf phloem cells, indicated a decrease in IAA loading into the phloem, which led to a decrease in the exudation of this hormone from the shoots of plants grown in low light conditions. A decrease in the delivery of IAA from the shoots resulted in a reduced content of this hormone in the roots and was accompanied by poor root branching. Since auxin is known to be required to induce lateral root growth, this low light effect is likely related to the long-distance signal from the leaves in the form of reduced auxin delivery through the phloem. A decrease in the abundance of the transcript of the *LAX3* gene involved in auxin transport in the roots probably contributes to the effect of low light on the branching of barley roots. 

## Figures and Tables

**Figure 1 biology-12-00787-f001:**
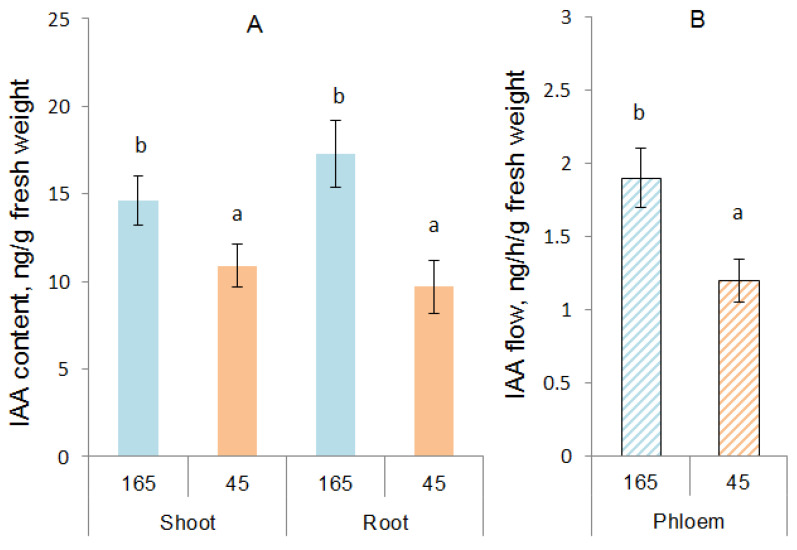
The content of IAA calculated per g of fresh weight of shoots and roots (**A**), and phloem flux of IAA (ng h^−1^ g^−1^ of leaf fresh weight) (**B**) of barley plants grown during one day at different illumination levels (165 vs. 45 µmol m^−2^ s^−1^ PAR). Statistically different means are marked with different letters (ANOVA, Duncan test at *p* < 0.05; *n* = 9).

**Figure 2 biology-12-00787-f002:**
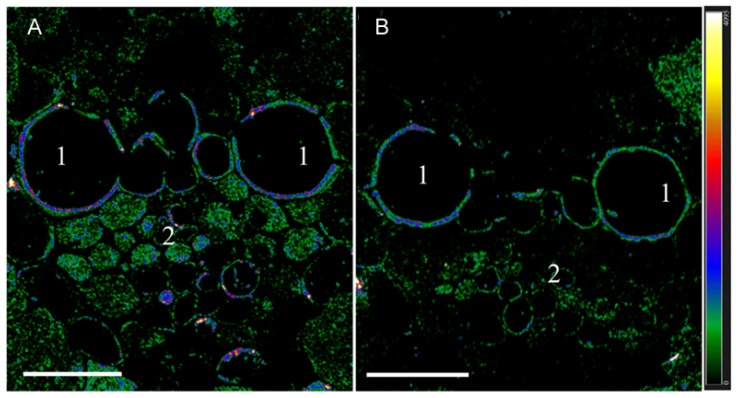
Immunohistochemical localization of IAA in vascular bundles of leaves of barley plants after their growing during one day at various levels of illumination. Images were taken from 10 independent sections per treatment and figure shows representative images. Intensity of fluorescence is displayed as a color-coded heatmap. (**A**)—control (165 µmol m^−2^ s^−1^), (**B**)—reduced illumination (45 µmol m^−2^ s^−1^). Scale bar 20 µm, 1—xylem, 2—phloem.

**Figure 3 biology-12-00787-f003:**
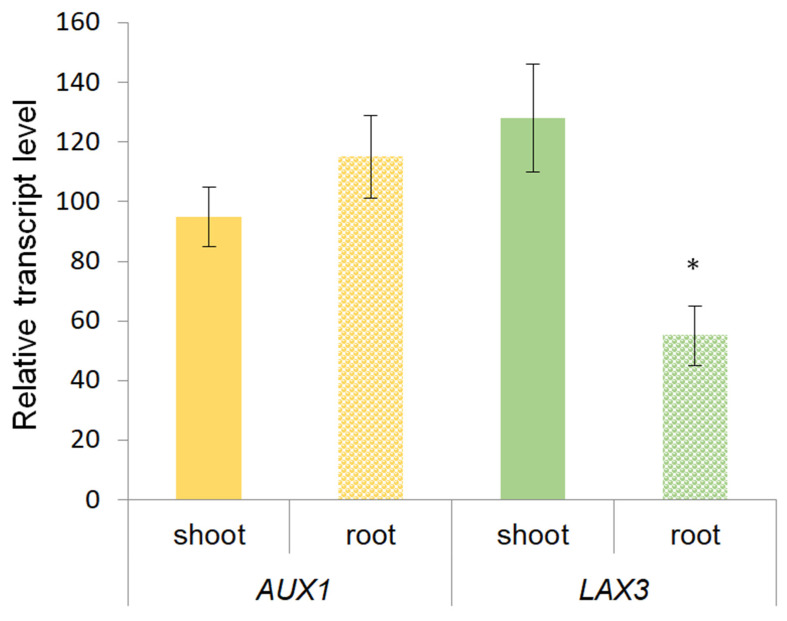
Relative level of *AUX1* and *LAX3* genes transcripts in barley plants grown during one day under low light (45 µmol m^−2^ s^−1^). Figure shows the data obtained under low light and presented as a percentage of control values obtained at illumination of 165 µmol m^−2^ s^−1^ (*n* = 9). Means for low light plants significantly different from the means for high-light plants are marked with asterisk.

**Table 1 biology-12-00787-t001:** List of primers used for quantitative RT-PCR analysis.

Genes	Chain	Primer Sequence 5′ to 3′	Registration Number in GenBank
*HvAUX1*	Forward	TCCGTGTAGCACACCATTACTT	XM_045121727.1
	Reverse	CAGGAATTTACTGTGCGATTGA	
*HvLAX3*	Forward	GGTCTCTAGTCGATCGGAAGG	XM_045107426.1
	Reverse	CCTCCTTCGGGTTACATTAGTT	
*HvACT*	Forward	GGTATACACGAAGCGACATACA	MK034133
	Reverse	GTAGAACCACCACTGAGAACAA	
*HvTubA*	Forward	GAGCGTCTCTCTGTTGACTATG	AK250165
	Reverse	TGGACAGGACACTGTTGTATG	
*HvGADPH*	Forward	GCCACTATTCTTCAGGGACTT	EF409626
	Reverse	CTTCTTGGCACCACCCTAATA	

**Table 2 biology-12-00787-t002:** The fresh biomass of shoots and roots, sum of length of all primary roots and lateral roots number of barley plants grown for 2.5 days under different illumination (165 and 45 µmol m^–2^ s^–1^ PAR (the Table shows means ± s.e., *n* = 40). The means for the two options were significantly different at *p* ≤ 0.05 (*t*-test).

Illumination, µmol m^−2^ s^−1^	Shoot Mass, mg	Root Mass, mg	Root Length, cm	Number of Lateral Roots
165	176 ± 3	80 ± 3	47.5 ± 1.9	28.0 ± 3.0
45	150 ± 4	56 ± 3	30.7 ± 1.7	2.8 ± 0.7

**Table 3 biology-12-00787-t003:** The content of ABA (ng) per g of fresh weight of shoot or root and phloem flux (ng h^−1^ g^−1^ of leaf fresh weight) of barley seedlings grown during one day at different illumination levels (*n* = 9).

	Shoot		Root		Phloem	
Illumination, µmol/m^2^/s	165	45	165	45	165	45
ABA content	4.0 ± 0.9	4.6 ± 1.0	11.1 ± 1.6	10.4 ±1.8	1.3 ± 0.2	1.6 ± 0.2

## Data Availability

Not applicable.

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
