# Peer review of "Effect of Low Light Stress on Distribution of Auxin (Indole-3-acetic Acid) between Shoot and Roots and Development of Lateral Roots in Barley Plants"

_biology, 2023, doi:10.3390/biology12060787_

Round 1

Reviewer 1 Report

The MS "Effect of low light stress on auxin distribution between shoot and roots and lateral root formation in barley plants" has an exciting theme and is an original and new contribution. Unfortunately, despite the good results, certain points need to be clarified before being approved for publication.

1. Abstract: Please include more significant results, a concluding remark, and the future scope of this study.

2. Introduction: Can be improved. Please provide more information about the selected crop, i.e., genetics, importance, yield, productivity, etc. Apart from this, a clear and concise objective is missing: why the author selected this topic, its scientific importance, research gaps, etc.

3. Materials and methods: Line 101, change "+40C for 24 Hours" to 40C for 24 Hours." Please make a separate subheading for "Purification and quantification of IAA and ABA". 

4. Line 186: Please change "statistics" to "statistical analysis."

5. Results: The presentation of results is very poor. It should be written in a logical manner and can be elaborated on in depth. PCR images are also missing.

Please clarify the role of the PIN protein and other transporters that are responsible for the transport of auxins and their polarity.

Please clarify the ARF, inhibitor of NPA, and role of LAX3 genes.

6. The conclusion is not written properly. It needs to be rewritten, avoiding unnecessary and irrelevant sentences. Apart from this, please add future investigations.

7. Botanical names (for example, Arabidopsis) should be in "ITALIC". Please correct it throughout the paper.

8. Please provide a list of abbreviations used in this study just before the reference list.

English Language and style is fine, however, some grammatical and punctuation errors observed while reviewing MS. In addition, some sentences are hard to understand. Therefore, the author should pay more attention to the usage of grammar and punctuation for the further improvement.

Author Response

Thanks to the respected reviewer for kind words about exciting theme and original and new contribution of our article. We carefully followed all valuable recommendation of reviewer.

  1. Abstract: Please include more significant results, a concluding remark, and the future scope of this study.

Response: According to this remark we added following sentences to the abstract: “Reduced content of IAA found in the plants under low light suggests inhibition of production of this hormone under these conditions” (significant result) “The results confirm importance of long distance transport of auxins for the control of the growth of roots under conditions of low light” (concluding remark) and “Further study of the mechanisms that control transport of auxins from shoots to roots in other plant species is required” (future scope of this study). At the beginning of the abstract we also added a sentence regarding the background of the problem: “Depending on their habitat conditions plants can greatly change the growth rate of their roots. However mechanisms of such responses remain insufficiently clear.”

  1. Introduction: Can be improved. Please provide more information about the selected crop, i.e., genetics, importance, yield, productivity, etc.

Response: In accordance we added to the sentence about genetics of barley (“Barley plants were chosen by us for the present research, since the genome of these plants is better studied than that of wheat plants, which facilitated the search for genes of interest”) one more sentence: “Barley is the fourth most important cereal crop after corn, rice and wheat in terms of production quantity. It is the predominant crop in some dryland agricultural systems (e.g., in Mediterranean area) [12].

Remark: Apart from this, a clear and concise objective is missing: why the author selected this topic, its scientific importance, research gaps, etc.

Response: We added to Introduction a sentence regarding the aim of the present research and its importance: “The aim of this research was to reveal involvement of auxin in the transfer of signals from shoot to root under low light, mechanisms of generation of this hormonal signal and its importance for the control of root branching. The mechanisms that ensure the exchange of signals between various plant organs are important for coordinating their growth and adaptation to a changing environment, and their deeper understanding should allow us to influence these processes in the direction necessary to increase plant productivity”. There are also sentences in the introduction about research gaps (“The mechanism of signal transduction from roots to shoots has been well studied… Signaling in the opposite direction from shoots to roots received less attention “).

  1. Materials and methods:Line 101, change "+40C for 24 Hours" to 40C for 24 Hours."

Response: It is “4°C for 24 hours”

- Please make a separate subheading for "Purification and quantification of IAA and ABA". 

Response: subheadings were introduced

  1. Line 186: Please change "statistics" to "statistical analysis."

Response: Statistics changed for statistical analysis

  1. Results: The presentation of results is very poor. It should be written in a logical manner and can be elaborated on in depth.

Response: To facilitate the logic of description of obtained results we wrote at the beginning of Result section that “Since our research was focused on the study of the mechanisms that control the growth response of roots to low light, at the first stage, the effects of low light on root growth associated with shoot growth responses were elucidated.” And before proceeding to describe the results of hormone assay we added: “To reveal involvement of hormones in the control of root growth responses to low light, concentration of IAA and ABA were determined at the next stage.”

  • PCR images are also missing.

Response: Unlike previous methods based on evaluating amplification products at the end of PCR (RT-PCR) by estimating the relative amount of transcripts from the images of their distribution in agarose, the method used in the present study (quantitative real-time PCR) is based on a different approach that does not require visualization of distribution of amplicons in agarose gel. When using this method, the amplicon content of each amplification cycle is expressed as a numerical value, from which curves or histograms are plotted, as we have done in Fig. 3. Our articles are published in many, including molecular biology journals, and at no time have the qRT-PCR results been required to be accompanied by any methodological Figures (e.g., Akhtyamova et al. published in Int. J. Mol. Sci. 2021, 22, 10680. doi.org/10.3390/ijms221910680 or Arkhipova et al. published in Microorganisms 2022, 10, 935. https://doi.org/10.3390/microorganisms10050935

  • Please clarify the role of the PIN protein and other transporters that are responsible for the transport of auxins and their polarity. Please clarify the ARF, inhibitor of NPA, and role of LAX3 genes.

Response: We are sorry that what we wrote in the introduction of the initial variant was obviously not sufficient. In the present variant, we specified everywhere that our work was focused on AUX1/LAR, which are auxin transporter so that to distinguish them from AUX/IAA-ARF (components of the auxin signaling pathway). We added more information about auxin transporters in result section: “Since in the previous experiments with wheat, expression of genes encoding PIP carriers responsible for auxin efflux [9] was not changed in shoots under low light [8], we focused on AUX1 and LAX3 genes encoding another class of carriers that transport auxins in the opposite direction (into cells) [10].” We also added more information about auxin transporters to the Discussion: “(AUX1/LAX) family members are the major auxin influx carriers whereas AUX1 and LAX3 both regulate lateral root development (Swarup et al., 2008). [12]. Lateral roots development has been shown to be dependent on the local expression of LAX genes around lateral root primordial, which facilitates lateral root emergence in Arabidopsis [33]. In this case, induction of the LAX3 gene increased auxins in the root cortex cells, which lead to up-regulation of cell wall remodeling enzymes to facilitate passage of primordia through the cortex. Therefore down-regulation of homologous barley gene found in our experiments in roots under low light may be the cause of inhibition of lateral root growth and is consistent with some literature data [34].” We also tried to clarify the sentence about NPA: “In these experiments the use of inhibitor of auxin transport (naphthylphthalamic acid, NPA) imitated the effects of shading and thereby confirmed that reduced level of root auxin in the presence of neighbors was associated with a decrease in the hormone transport from the shoot.”

  1. The conclusion is not written properly. It needs to be rewritten, avoiding unnecessary and irrelevant sentences. Apart from this, please add future investigations.

Response: Conclusion was rewritten: “Thus, data obtained in the present work indicate a decrease in the concentration of IAA in the shoots, leaves and phloem sap of barley plants exposed to low light intensity, which suggests a decrease in the synthesis of auxin in the leaves under the influence of this external factor. A decrease in auxin concentration, detected by immunolocalization in leaf phloem cells, indicated a decrease in IAA loading into phloem, which led to a decrease in exudation of this hormone from shoots of plants grown in low light conditions. A decrease in delivery of IAA from the shoots resulted in reduced content of this hormone in the roots and was accompanied by poor root branching. Since auxin is known to be required to induce lateral root growth, this low light effect is likely related to the long-distance signal from the leaves in the form of reduced auxin delivery through the phloem. A decrease in abundace of the transcript of the LAX3 gene involved in auxin transport in the roots probably contributes to the effect of low light on the branching of barley roots.

  1. Botanical names (for example, Arabidopsis) should be in "ITALIC". Please correct it throughout the paper.

Response: Arabidopsis is changed for Italic throughout the text

  1. Please provide a list of abbreviations used in this study just before the reference list
  2. Response: The list is provided

Reviewer 2 Report

The manuscript by Korobova et al. titled, “Effect of low light stress on auxin distribution between shoot and roots and lateral root formation in barley plants” investigated how barley plant (Hordeum vulgare L., cv. Prairie) growth was affected by low illumination i.e., at 45 micromol m-2 s-1 photosynthetically active radiation (PAR) compared to 165 micromol m-2 s-1 PAR by monitoring shoot and root mass, root length and the number of lateral roots. Plant hormone levels, IAA and ABA, were monitored by enzyme and hormone immunoassays and localization studies done on leaf samples. In addition, RNA transcripts levels of HvAUX1 and HvLAX3 were measured in the shoots and roots due to AUX1 being involved in auxin transport over long distances from the shoot to the roots via the vascular bundle, while LAX (or LAX3) is involved in maintaining local gradients.

The authors observed that after one day low illumination treatment a decrease in shoot and root mass, a decrease in root length as well as the number of laterals.  IAA content and flow also decreased. However, it appeared that there was no significant change with ABA content and flow. IAA immunolocalization intensity in the vascular leaf bundles appeared to be lower in a reduced illumination samples compared to a control sample. They also observed that LAX transcript level in the roots of plants grown under low light conditions was lower compared to control plants.

The authors then concluded that changes in the delivery of IAA from shoots to roots was responsible for root branching under low light.

Here are some suggestions:

Did the authors try to supplement IAA to the roots under low light conditions to observe if this would affect root mass and lateral root number and alleviate the negative effects of low light condition? Or excising the leaves and supplementing IAA from above?

Line 133- Please include a sub-heading related to purification and quantification of IAA and ABA in shoots and roots.

Line 160 – space or parentheses around ‘pH5.5 : 3%....’

Table 2: are shoot and root mass, fresh weight or dry weight – please clarify in the method and legend of table 2.

Figure 2. is a representive of how many samples observed? Also would it be possible to get range of intensity signal level in relation to the colours? Figure 2 is not referred to in the result section (only Figure 1 and 3). Were any immunolocalization work done in the root section for comparison?

Line 168 mentioned about ABA immunolocalization. Data missing.

Figure 3: please correct the AUX1 label in the graph and please state which one is under low and control light conditions? Is the yellow and green correlate to this? Please write an explanation in legend to remove any ambiguity or confusion.

Also has there been any immunolocalization work done on barley leaves with AUX1 or LAX? Could this be important for future work or discussion.

Author Response

We are most grateful to the reviewer for careful analysis of our work and valuable comments. We addressed all of them.

  1. Did the authors try to supplement IAA to the roots under low light conditions to observe if this would affect root mass and lateral root number and alleviate the negative effects of low light condition? Or excising the leaves and supplementing IAA from above?

Response: This is really important and can serve an object of further study. In accordance we added to the text; “It would be interesting to know whether applying IAA to barley plants under low light can restore root branching under low light, which could be a subject for further study.”

  1. Line 133- Please include a sub-heading related to purification and quantification of IAA and ABA in shoots and roots.

Response: subheading “Hormone purification” and “Hormone quantifications” were included

  1. Line 160 – space or parentheses around ‘pH5.5 : 3%....’

Response: This paragraph was shortened and rewritten to avoid similarities with the previously published description of the method. In the revised version this sentence is absent

  1. Table 2: are shoot and root mass, fresh weight or dry weight – please clarify in the method and legend of table 2.

Response: Sorry for this inconsistency. Table legend is modified and mentions fresh mass of shoot and root

  1. Figure 2. is a representive of how many samples observed? Also would it be possible to get range of intensity signal level in relation to the colours? Figure 2 is not referred to in the result section (only Figure 1 and 3).

Response: Thanks for this remark. In accordance with the first sentence we added to the figure legend that “Images were taken from 10 independent sections per treatment and figure shows representative images.” We are sorry that description of Figure 2 was too short and therefore not noticed by reviewer. So we extended description of this figure and added description of color-coded intensity of fluorescence. The sentences are as follows: “Under low light fluorescence was present in fewer cells of phloem than in the control. Fluorescence of phloem cells was mainly coded in green indicating a low level of auxin content, while in the control, red and blue color indicated higher auxin content according to the heat map.”

  • Were any immunolocalization work done in the root section for comparison?
  • Response: We performed immunolocalization of IAA in root primordia, but found no difference between plants grown under low and normal illumination. According to the remark of the reviewere we added the results into Supplementary material and their description into the Result : “There were no differences in the fluorescence of root promordia between plants grown at different illumination levels (Figure 1S). The small supply of IAA to the roots under low illumination was apparently still sufficient to provide the reduced number of lateral roots with auxin.”).
  1. Line 168 mentioned about ABA immunolocalization. Data missing.

Response: As mentioned above, description of M & M was greatly modified and ABA immunolocalization is no longer mentioned (this was a mistake)

  1. Figure 3: please correct the AUX1 label in the graph and please state which one is under low and control light conditions? Is the yellow and green correlate to this? Please write an explanation in legend to remove any ambiguity or confusion.

Response: AUX1 label was corrected. We are sorry for unclear description. The figure presents data for low light expressed as percent of data for control (high light). We tried to make it clearer and added: “Figure shows the data obtained under low light and presented as a percentage of control values obtained at illumination of 165 µmol m–2 s–1 (n=9).”

  1. Also has there been any immunolocalization work done on barley leaves with AUX1 or LAX? Could this be important for future work or discussion.

Response: It would be great to have these data. Unfortunately we have no antibodies against AUX1 or LAX of barley.

Reviewer 3 Report

The materials and methods should be highlighted more clearly.

The results also should be more clearly presented (editing), because it is a one part difficult to read.

Moreover, the practical aspect of research should be presented althought the work is basic research.

Author Response

We are most grateful to the reviewer for careful analysis of our work and valuable comments. We addressed all of them

  1. The materials and methods should be highlighted more clearly.

Response: We rewrote M & M section and divided into subsections in hope make it clearer

  1. The results also should be more clearly presented (editing), because it is a one part difficult to read.

Response. To facilitate the logic of description of obtained results we wrote at the beginning of Result section that “Since our research was focused on the study of the mechanisms that control the growth response of roots to low light, at the first stage, the effects of low light on root growth associated with shoot growth responses was elucidated.” And before proceeding to describe the results of hormone assay we added: “To reveal involvement of hormones in the control of growth root growth responses to low light, concentration of IAA and ABA were determined at the next stage”. We extended description of immunolocalization data: “Under low light fluorescence was present in fewer cells of phloem than in the control. Fluorescence of phloem cells was mainly coded in green indicating a low level of auxin content, while in the control, red and blue color indicated higher auxin content according to the heat map.” We specified in the legend of Figure 2 that “Images were taken from 10 independent sections per treatment and figure shows representative images.” We also tried to explain clearer, what Figure 3 shows and wrote: “Figure shows the data obtained under low light and presented as a percentage of control values obtained at illumination of 165 µmol m–2 s–1 (n=9).” We also added justification of the study of expression of the genes: “. Since in the previous experiments with wheat, expression of genes encoding PIP carriers responsible for auxin efflux [9] was not changed in shoots under low light [8], we focused on AUX and LAX genes encoding another class of carriers that transport auxins in the opposite direction (into cells) [10].”

  1. Moreover, the practical aspect of research should be presented although the work is basic research.

Response: In accordance with this remark we added to the end of Introduction: “The mechanisms that ensure the exchange of signals between various plant organs are important for coordinating their growth and adaptation to a changing environment, and their deeper understanding should allow us to influence these processes in the direction necessary to increase plant productivity.”

Round 2

Reviewer 1 Report

The authors have not taken all the suggestions/comments seriously while revising the manuscript such as: they did not provide numerical data in the abstract, PCR images in the results which suspected its validity and authenticity (they responded with arguments for not giving PCR images "Unlike previous methods based on evaluating amplification products at the end of PCR (RT-PCR) by estimating the relative amount of transcripts from the images of their distribution in agarose, the method used in the present study (quantitative real-time PCR) is based on a different approach that does not require visualization of distribution of amplicons in agarose gel. When using this method, the amplicon content of each amplification cycle is expressed as a numerical value, from which curves or histograms are plotted, as we have done in Fig. 3. Our articles are published in many, including molecular biology journals, and at no time have the qRT-PCR results been required to be accompanied by any methodological Figures (e.g., Akhtyamova et al. published in Int. J. Mol. Sci. 2021, 22, 10680. doi.org/10.3390/ijms221910680 or Arkhipova et al. published in Microorganisms 2022, 10, 935. https://doi.org/10.3390/microorganisms10050935s).

The authors have not taken all the suggestions/comments seriously while revising the manuscript such as: they did not provide numerical data in the abstract, PCR images in the results which suspected its validity and authenticity (they responded with bad arguments like some of their papers previously published without PCR images).

Author Response

It seems to us that the reviewer is referring to RT-PCR (reverse-transcription polymerase chain reaction) and not the modern method that we use: Quantitative real-time reverse-transcription polymerase chain reaction (qRT-PCR).

The RT-PCR method has previously been used to determine the relative transcript content of individual genes. In this method, after cDNA synthesis, amplification with specific primers is carried out and the amplification product is evaluated only once at the end of PCR. Indeed, in this approach, PCR products are electrophoresed in agarose gel, fragments are stained and scanned to determine the luminescence intensity of the DNA fragments. In this method of assessing relative transcript content, photography of the agarose gene is highly desirable.

In the current method (qRT-PCR), RNA isolation and cDNA synthesis are performed in exactly the same way as previously in the RT-PCR method. However, thanks to new equipment, for example, the QuantStudio™5 Real-Time PCR System by Thermo Fisher Scientific (Singapore), which we used and the new approach, the amplicon content is detected not only at the end of the last PCR cycle but at every PCR cycle, and these values are immediately displayed in numerical values which do not require any graphic illustration. This allows one to observe the dynamics of cDNA amplification and, based on these results, a numerical value of the amplicon content, which reflects the level of cDNA used for amplification, which in turn corresponds in numerical form to the content of the transcripts of the genes under analysis. Based on the numerical values obtained, curves or histograms reflecting the content of the transcripts are plotted, as presented in our manuscript in Fig. 3.

No additional illustrations are required for the modern method of analysis we used. Validation of qPCR is based on  curve analysis method which includes estimation of the efficiency of the PCR reaction (the fold increase of product per cycle).

Reviewer 2 Report

The modifications made in this revised version  greatly improved the manuscript. 

Suggestions:

Please make sure that the supplementary figure S1 is included with the legend as it was missing.

Author Response

Thank you

Round 3

Reviewer 1 Report

I agree with the author's response and justification. In the revised paper the authors have tried to revise the manuscript according to the comments/suggestions provided; however, some minor errors are detected which I listed below. The manuscript can be accepted after incorporating the following input: Comment 1. Line no. 206 Please change "The fresh mass" to "The fresh biomass"  Comment 2. Some references are not formatted according to the author's guidelines of "Biology" (For example in ref 1 line no. 407, the journal name is written as "J Exp Bot." which should be "J. Exp. Bot.". Similar problems are also in ref no. 2, 5, 7, 11, 12 (not abbreviated), 18,19, 22, 27 (not abbreviated), 29 (NOT in ITALIC Font), 32, 34. Please recheck and strictly follow the "Biology" Journal guidelines for references.

Author Response

We are most grateful to the respected reviewer for careful analysis of our article and valuable comments. The article was revised according to all of them

Comment 1. Line no. 206 Please change "The fresh mass" to "The fresh biomass" 

Response: “bio” was added to “mass”

Сomment 2. Some references are not formatted according to the author's guidelines of "Biology" (For example in ref 1 line no. 407, the journal name is written as "J Exp Bot." which should be "J. Exp. Bot.". Similar problems are also in ref no. 2, 5, 7, 11, 12 (not abbreviated), 18,19, 22, 27 (not abbreviated), 29 (NOT in ITALIC Font), 32, 34. Please recheck and strictly follow the "Biology" Journal guidelines for references.

Response: Sorry for so many mistakes on our part! We corrected the list.